# Marketplace shrimp mislabeling in North Carolina

**Morgan L. Korzik**[1], **Hannah M. Austin**[1], **Brittany Cooper**[1], **Caroline Jasperse**[1], **Grace Tan**[1], **Emilie Richards**[1], **Erin T. Spencer**[2], **Blaire Steinwand**[1], **F. Joel Fodrie**[3], **John F. Bruno**[1]*

**1** The Department of Biology, The University of North Carolina at Chapel Hill, Chapel Hill, North Carolina, United States of America, **2** Environment, Ecology, and Energy Program, The University of North Carolina at Chapel Hill, Chapel Hill, North Carolina, United States of America, **3** Institute of Marine Sciences, The University of North Carolina at Chapel Hill, Morehead City, North Carolina, United States of America

* jbruno@unc.edu

**Data Availability Statement:** Data can be found publicly at the following Figshare pages: https://figshare.com/articles/Marketplace_Shrimp_Mislabeling_in_North_Carolina/9505247 https://

## Abstract

Seafood mislabeling occurs in a wide range of seafood products worldwide, resulting in public distrust, economic fraud, and health risks for consumers. We quantified the extent of shrimp mislabeling in coastal and inland North Carolina. We used standard DNA barcoding procedures to determine the species identity of 106 shrimp sold as "local" by 60 vendors across North Carolina. Thirty-four percent of the purchased shrimp was mislabeled, and surprisingly the percentage did not differ significantly between coastal and inland counties. One third of product incorrectly marketed as "local" was in fact whiteleg shrimp: an imported and globally farmed species native to the eastern Pacific, not found in North Carolina waters. In addition to the negative ecosystem consequences of shrimp farming (e.g., the loss of mangrove forests and the coastal buffering they provide), North Carolina fishers—as with local fishers elsewhere—are negatively impacted when vendors label farmed, frozen, and imported shrimp as local, fresh, and wild-caught.

## Introduction

Shrimp is the most popular seafood in the United States. Shrimp represents a quarter of America's annual per capita seafood consumption and the average American eats about five pounds of shrimp each year [1]. This results in over one billion pounds of shrimp consumed annually in the United States alone, 80% of which is imported [1]. In 2018, the U.S. imported 136 million pounds of shrimp, primarily from Indonesia, India, and Ecuador, which accounted for 33% of all reported seafood imports [2,1].

Shrimping has deep cultural and economic roots along the North Carolina coast. In 2017, commercial fishers caught 13.9 million pounds of shrimp, which is 82.9% greater than the previous five year average [3]. That same year, shrimp was the highest earning fishery in the state, valuing $29.6 million, exceeded in catch weight only by blue crab [3]. Despite a 9% annual decline in commercial landings for all species in North Carolina in 2017, the value of local shrimp was at an all-time high, suggesting sustained demand. However, pressure from

figshare.com/articles/Shrimp_sequences_and_
accession_numbers_for_Korzik_et_al_2020/
11865969.

**Funding:** This study was supported by the
Department of Biology at The University of North
Carolina at Chapel Hill and funded by the QEP
(Quality Enhancement Plan) CURE (Course-based
Undergraduate Research Experience) initiative. The
project was also partially funded by the National
Science Foundation (OCE #1737071 to JB).

**Competing interests:** The authors have declared
that no competing interests exist.

regulations and imported seafood products are believed to be at least partially responsible for longer-term decline in North Carolina's seafood industry [4]. The number of licensed commercial fishermen in North Carolina declined 41% between 1995 and 2011, and the number of seafood processors on the eastern shore of the state declined 36% between 2000 and 2011 [4,5]. Despite the increasing prevalence of imported seafood, 92% of North Carolina consumers surveyed by NC Sea Grant indicated they prefer to eat local seafood over imported seafood [4].

In response to increasing pressure on commercial fisheries from seafood imports, organizations in North Carolina and other states have conducted outreach to consumers encouraging them to "eat local seafood". For example, NC Catch is a local seafood advocacy organization in North Carolina that provides information about vendors that sell local seafood [6]. The intent is to help consumers to make informed decisions regarding seafood purchases that support the local fishing industry [6]. However, this is only effective when seafood products are accurately labeled.

For this study, we defined seafood mislabeling as occurring when a species was substituted with another type of seafood, including varieties of lower economic value [7]. Commercial catch restrictions on in-demand species can create an economic incentive for vendors to sell lower-valued fish as more expensive ones [7]. Seafood mislabeling can occur at any point along the supply chain, from initial harvest to consumer purchase. It can be difficult to determine where in the supply chain mislabeling has occurred, allowing the practice to continue despite growing public awareness [8].

According to a 2019 study, 21% of seafood is mislabeled in the United States, and 8% is mislabeled worldwide [9,10]. However, mislabeling rates vary enormously among species and seafood types, making generalizations impossible [10]. Mislabeling has a myriad of potential consequences including exacerbating over-fishing, negative impacts on human health, and perpetuating human rights abuses in international fisheries [11,12,13]. As seafood products can be difficult to distinguish visually, an increasing number of studies are using DNA barcoding to quantify the frequency of mislabeling across different species and geographic regions [14].

We used standard DNA barcoding techniques to quantify the level of shrimp mislabeling in North Carolina. Few studies have focused specifically on shrimp mislabeling, and none have assessed the prevalence of shrimp mislabeling in North Carolina. We considered three shrimp species, *Farfantepenaeus aztecus*, *Litopenaeus setiferus*, and *Farfantepenaeus duorarum*, as "local" to North Carolina. *F. aztecus*, or brown shrimp, is the most abundant shrimp species in North Carolina and accounts for 67% of the state's shrimp catch [15]. *L. setiferus*, or white shrimp, is the second-most abundant shrimp species and accounts for approximately 28% of shrimp landings [15]. *F. duorarum*, or pink shrimp, only account for 5% of the state's shrimp catch [15].

## Materials and methods

To determine the frequency of shrimp mislabeling in North Carolina, we collected "local" shrimp sold at coastal and inland vendors, including grocery stores and seafood-specific markets. All vendors reported to NC Catch that they sold seafood caught in North Carolina. Some shrimp had signage that explicitly labeled them as local, while others were verified as local by the retail personnel. Samples were only collected when the vendor explicitly or verbally confirmed the product was local or North Carolina shrimp. We purchased shrimp from 60 vendors across North Carolina (31 were inland vendors and 29 were coastal) in spring (May 18 through June 5) and fall (September 20 through November 4) of 2017 and during the spring of 2018 (May 18 through June 2) (Fig 1). We defined inland vendors as ones located in a land-

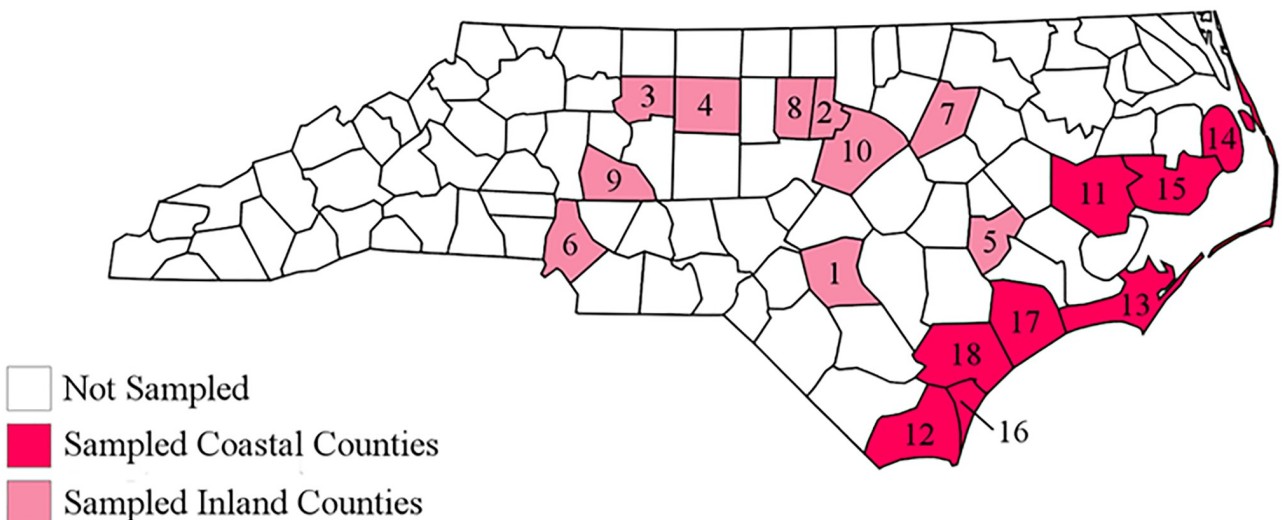

**Fig 1. Distribution of sampled vendors in North Carolina.** The inland counties include Cumberland (1), Durham (2), Forsyth (3), Guilford (4), Lenoir (5), Mecklenburg (6), Nash (7), Orange (8), Rowan (9), and Wake (10). The coastal counties include Beaufort (11), Brunswick (12), Carteret (13), Dare (14), Hyde (15), New Hanover (16), Onslow (17), and Pender (18).

locked county. The cost ($/lb) for each sample collected was recorded. Twenty-three of the 60 vendors were sampled in multiple years and/or seasons, i.e., to quantify labeling consistency. For 10 vendors, we sampled more than one "variety" of shrimp on a given sampling date. Such varieties (e.g., of different sizes, deveined or not, etc.) were typically displayed in separate containers, usually a metal pan. For every sampling visit, we obtained three separate shrimp from each sampled variety and froze these in 2-mL plastic screw cap tubes. For most of the vendors sampled in the spring of 2017, we processed multiple shrimp to test whether there was a mixture of mis- and correctly-labeled products of a given variety, mixed together in a single display container. We never found any variance among replicate shrimp (i.e., sample identity was always concordant), so we did not continue to process every replicate for samples collected in the fall of 2017 and in 2018. The exception was when we were not able to successfully sequence the first sample.

Following the DNEasy extraction protocol (Qiagen, INC), we extracted genomic DNA from approximately 20mg shrimp tissue. To identify individual samples to the species-level we focused on sequencing the mitochondrial DNA cytochrome oxidase I gene (CO1). This gene is well-conserved, has little variation within a species, and has enough variation between species to make it a good candidate for our study. It has been used in other seafood mislabeling studies (e.g., [11,16,14]). We amplified CO1 sequences from extracted DNA following the PCR protocol outlined in Willette et al. [14] and a primer cocktail from Ivanova et al. ([17]; primer set C_FishF1t1–C_FishR1t1). To prepare a 25μL sample for PCR, we combined the DNA with a primer cocktail of CO1_F1, CO1_F2, CO1_R1, and CO1_R2, deionized water, and a PuRe Taq Ready-To-Go PCR bead containing the necessary PCR components. In the thermal cycler, the samples went under 35 cycles of 95°C for denaturation, 50°C for annealing, and 70°C for extension. A negative control containing all of the PCR components except DNA was used to test for contamination. We ran the PCR products on a 1% agarose gel to determine whether PCR amplification of the DNA was successful. Samples with successful ~650 base pair bands were sent to an ETON Bioscience facility in Raleigh, NC for sequencing. Chromatograms of successfully sequenced regions were then matched against CO1 sequences of known samples on National Center for Biotechnology Information's nucleotide collection database GenBank

using the Basic Local Alignment Search Tool (BLAST). We only concluded the identity of a species if the percent identity and query coverage was greater than or equal to 98% and the e-value was close to zero. Additionally, we recorded the percent identity of the next species down in the BLAST results. Samples identified as *Litopenaeus setiferus* were considered to be local North Carolina shrimp while *Litopenaeus vannamei* samples were determined to be Pacific whiteleg shrimp, and thus mislabeled.

## Results

Of 128 samples collected, 106 were successfully sequenced: 47 were from inland vendors, and 59 from coastal vendors. Thirty-six (34%) were identified as *Litopenaeus vannamei* and 70 were *Litopenaeus setiferus*. These were the only two shrimp species identified in our study. *L. setiferus* or "white shrimp" are native to the western Atlantic and are harvested along the eastern coast of the United States, and in the Gulf of Mexico. *L. vannamei*, whiteleg shrimp, does not occur in the western Atlantic. Shrimp sold as "local" and identified as *L. vannamei* were considered mislabeled (and were assumed to be imported, not fresh, and likely farmed).

There was no statistical difference in mislabeling frequency between coastal and inland vendors ($\chi^2$ = 0.212 and p = 0.65). 35% of vendors mislabeled local shrimp at least once. Of the ten resampled vendors, six sold both correctly labeled and mislabeled shrimp. The price of mislabeled shrimp (mean: $11.00/lb) was significantly lower than that of the correctly labeled samples (mean: $13.20/lb, p-value = 0.001, t-test, Fig 2).

## Discussion

Of the 60 sampled vendors across North Carolina, 35% substituted imported shrimp for local shrimp at least once. This statewide mislabeling frequency is consistent with the 35% shrimp mislabeling frequency nationwide [18]. Although this frequency is lower than that of other species in North Carolina, e.g., red snapper, [19] the results suggest shrimp mislabeling is a fairly common problem. Interestingly, the average selling price of accurately labeled shrimp was more than $2 greater, suggesting that vendors knowingly or subconsciously placed lower values on mislabeled products.

Shrimp mislabeling has ecological, economic, and human health impacts. Locally-caught white shrimp is a smart seafood choice by the National Oceanic and Atmospheric Association (NOAA) because it is sustainably harvested and managed in the South Atlantic and Gulf of Mexico [20]. White shrimp populations are above target levels, and gear restrictions, such as the required inclusion of turtle excluder devices and bycatch reduction devices, are in place to minimize impacts of trawling on benthic ecosystems [20].

Pacific whiteleg shrimp are harvested through trawling, which can be destructive to benthic ecosystems and result in high levels of bycatch without gear restrictions like those present in the United States [21,22]. Additionally, Pacific whiteleg shrimp is the most widely farmed shrimp species in the world and is cultivated in at least 27 countries [23]. Shrimp farming poses a number of environmental risks, including mangrove destruction and the associated loss of native biodiversity and ecosystem services [24]. Shrimp farms often use large doses of antibiotics to prevent the spread of disease, which can contribute to antibiotic resistance in both shrimp and human populations [25]. Some antibiotics used to treat farmed shrimp, such as enrofloxacin and chloramphenicol, are not advised for human use due to risks of cancer and immune system damage [25]. A study in Thailand found 74% of interviewed shrimp farmers used up to 13 different antibiotics in their shrimp ponds, sometimes daily, and many were poorly informed about safe application of antibiotics [26].

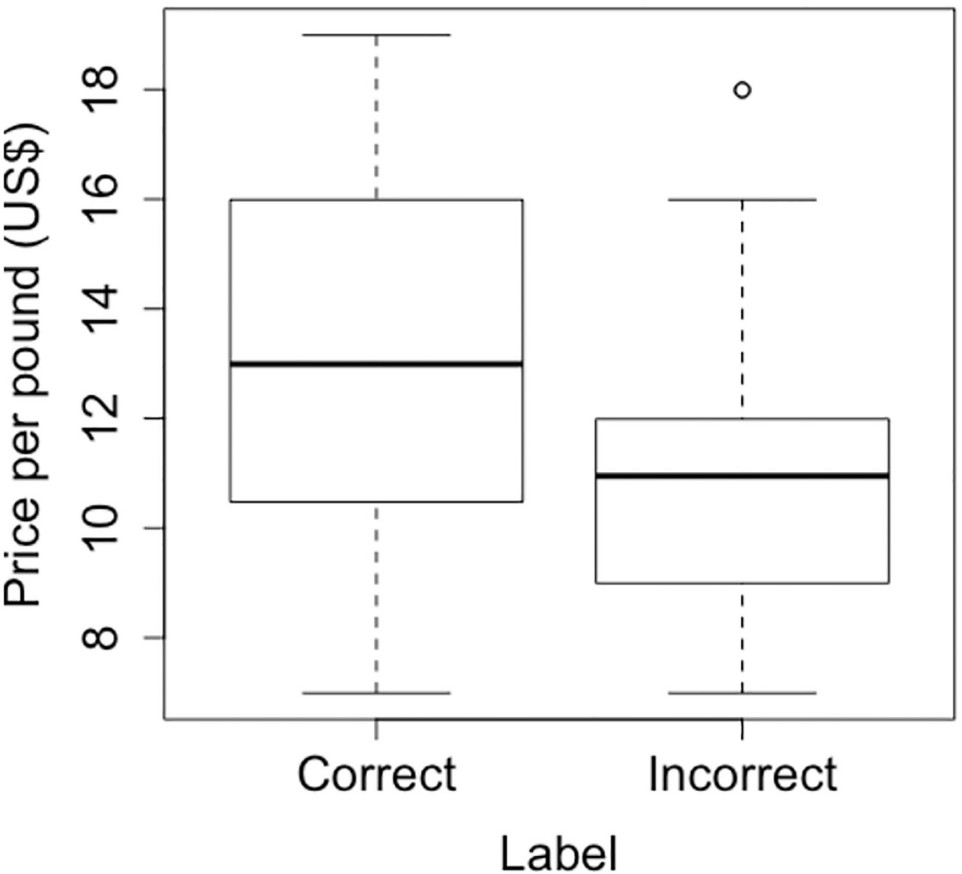

**Fig 2. Price of correctly labeled and mislabeled shrimp purchased in North Carolina.** Boxplots of the distribution of price per pound of all shrimp that were found to be correctly labeled as "local" compared the price of shrimp incorrectly labeled as "local". The difference was statistically significant (p-value = 0.001, t-test).

There are also human rights concerns with imported shrimp [13]. In 2014, multiple news organizations reported slavery practices on Thai fishing vessels harvesting offshore fish to use for farmed shrimp feed [27]. Exposure of these practices led to a consumer movement to eat seafood that was both environmentally sustainable and ethically harvested. Mislabeling imported shrimp as locally-caught shrimp undermines the power of the consumer to spend their money in a way that aligns with their values. This also fosters consumer distrust in the seafood industry, which could lead to decreased spending on seafood products.

Surprisingly there was no difference in mislabeling frequency between coastal and inland vendors. Of the 23 vendors sampled in both 2017 and 2018, six sold both correctly-labeled and mislabeled shrimp across different years. Further research is needed to determine whether there are temporal trends in mislabeling: revisiting vendors throughout the year could help determine whether mislabeling frequency changed based on seasonal fishery closures or tourism activity. Mislabeling frequency could be higher when market demand is high, yet the commercial shrimp fishery is closed or when the shrimp are out of season. It would also be useful to determine where along the seafood supply chain most mislabeling occurs.

Increasing public awareness of mislabeling is an important initial step towards reducing its prevalence. Instituting positive and negative incentives within the industry to promote accurate labeling could also help. For example, states such as Florida and Louisiana have passed

legislation penalizing vendors that label specific seafood products (e.g., shrimp and grouper) inaccurately or vaguely. The development of independent certification (e.g., by a third-party NGO) for labeling accuracy, rewarding careful and honest vendors and distributors, could also provide the market incentives needed to overcome the benefits of mislabeling (such as greater profit margin).

## Acknowledgments

We thank Kelly Hogan and Interim Chancellor Kevin Guskiewicz for encouraging Bruno and Steinwand to develop the Seafood Forensics class. We thank Christopher Martin, Chris Willett, and Sabrina Burmeister for sharing their lab space and equipment with us. We also thank the many students who took the class and collected and processed samples. From the spring 2017 class, we thank Moza Hamud, Meredith McNairy, and Rachel Peterson. From the summer 2017 class, we thank Aravindhkrishna Ajithkumar, Alanna Dai, Jonathan Dolan, Justin Freeman, Saidou Jallow, Hyun Kim, Matthew Logan, Assem Patel, and Zachary Young. From the fall 2017 class, we thank Hanan Alazzam, Alexandra Barry, Christina D'Ovidio, Daniel Efird, Aaron Friedman, Tate Giddens, Mike Grossi, Amaya Martinez, Baily Mcinnes, Kirsi Oldenburg, Steve Park, Farhin Shaikh, Grace Steinman, Nicolas Tobar, Bren Woods. And from the summer 2018 class, we recognize Hannah Austin, Jasmine Barnes, Kane Cooper, Julianna Evans, Cassidy Manzonelli, Megan Ochs, and Carlos Urquilla. We also thank Sandy Bruno for collecting samples from New Bern, North Carolina and Ann Simpson of NC Catch for her ongoing guidance and for editing the manuscript. Additionally, we thank Paul Gabrielson for sharing his expertise and helping us with the DNA analysis and submitting the sequences to GenBank.

## Author Contributions

**Conceptualization:** F. Joel Fodrie.

**Data curation:** Morgan L. Korzik, Hannah M. Austin, Brittany Cooper, Caroline Jasperse, Grace Tan, Emilie Richards, Erin T. Spencer.

**Funding acquisition:** John F. Bruno.

**Methodology:** Blaire Steinwand, John F. Bruno.

**Software:** Emilie Richards.

**Supervision:** Blaire Steinwand, John F. Bruno.

**Writing – original draft:** Morgan L. Korzik, Hannah M. Austin, Erin T. Spencer, John F. Bruno.

**Writing – review & editing:** Morgan L. Korzik, Hannah M. Austin, Brittany Cooper, Caroline Jasperse, Grace Tan, Emilie Richards, Erin T. Spencer, Blaire Steinwand, F. Joel Fodrie, John F. Bruno.

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
