## [Decision Letter · Decision Letter 0]

26 Sep 2019

PONE-D-19-24180

Marketplace Shrimp Mislabeling in North Carolina

PLOS ONE

Dear Dr. Bruno,

Thank you for submitting your manuscript to PLOS ONE. After careful consideration, we feel that it has merit but does not fully meet PLOS ONE’s publication criteria as it currently stands. Therefore, we invite you to submit a revised version of the manuscript that addresses the points raised during the review process.

I found this to be a well written and interesting manuscript and both reviewers agreed. Overall, the text was clear and concise and the results timely. Both reviewers have provided relatively minor comments to improve the clarity of the paper and I agree. I have some generally minor editorial and formatting comments that need to be addressed. I do think that the methods need to be spelled out a bit more and the discussion could do a bit more to clarify the ramifications of the results. But I think these changes can be made pretty easily and both reviewers have provided detailed comments to assist the authors in their revision. I strongly encourage the authors to consider and address all the comments provided.

We would appreciate receiving your revised manuscript by Nov 10 2019 11:59PM. To enhance the reproducibility of your results, we recommend that if applicable you deposit your laboratory protocols in protocols.io, where a protocol can be assigned its own identifier (DOI) such that it can be cited independently in the future. For instructions see: http://journals.plos.org/plosone/s/submission-guidelines#loc-laboratory-protocols

We look forward to receiving your revised manuscript.

Kind regards,

Heather M. Patterson, Ph.D.

Academic Editor

PLOS ONE

Journal Requirements:

2. In your Methods section, please provide additional details regarding the shrimps used in your study and ensure you have described the source. For more information regarding PLOS' policy on materials sharing and reporting, see https://journals.plos.org/plosone/s/materials-and-software-sharing#loc-sharing-materials.

Reviewers' comments:

Reviewer's Responses to Questions

**Comments to the Author**

1. Is the manuscript technically sound, and do the data support the conclusions?

Reviewer #1: Yes

Reviewer #2: Yes

2. Has the statistical analysis been performed appropriately and rigorously? 

Reviewer #1: Yes

Reviewer #2: I Don't Know

3. Have the authors made all data underlying the findings in their manuscript fully available?

Reviewer #1: No

Reviewer #2: Yes

4. Is the manuscript presented in an intelligible fashion and written in standard English?

Reviewer #1: Yes

Reviewer #2: Yes

5. Review Comments to the Author

Reviewer #1: This is a very well-written and interesting paper for which I have little to say or add. I think it makes a great contribution to the literature on mislabeling, in fact, I haven’t seen an empirical paper as useful and well-framed in this field for some time.

The most important comment I have is that reporting only the top Blast hit isn’t convincing enough (see below). What follows are some specific comments organized by line number.

Lines 38-39: Why “Despite” - seems vague, perhaps implying that the domestic market could supply domestic demand. I doubt that, but the manuscript should more clearly establish that if it’s true or tweak this sentence.

Lines 48-52: This seems likely, but I suppose this suggests a causal relationship that isn’t necessarily there. What other factors might control declines in landings, licenses, and processors? Regulations?

Lines 70-71: Oceana has a more recent study: Warner, K., Roberts, W., Mustain, P., Lowell, B., & Swain, M. (2019). Casting a Wider Net: More Action Needed to Stop Seafood Fraud in the United States, (March), 19. https://usa.oceana.org/publications/reports/casting-wider-net-more-action-needed-stop-seafood-fraud-united-states

Line 93: How was certification by NC Catch determined by the authors? What I’m wondering is if the claim itself might often be fraudulent.

Line 111: Qiagen misspelled.

Line 119: I looked at Willette and Ivanova and I don't understand the use of 4 primers (those papers do not incude them). Were different pairs of primers used on different specimens or was this some sort of nested PCR?

Lines 128-130: The Blast results can be more convincing if something like the top X hits were all the same species and the next best hit that was not that species was <96% or something like that. There are many mistakes in GenBank, so suppose a top hit was misidentified by whoever submitted the sequence. Putting the sequences in a phylogeny can be helpful to, to see that the unknowns sit in monophyletic clades that are all the same species. BOLD might be worth checking because one can have much more confidence in the ID there.

Line 139: I would be as explicit as possible about what is meant by “native.” “Does not occur in the Western Atlantic” or “has never been reported from the Western Atlantic…”

Line 146: Is each sample truly independent? If two were bought on the same day from the same vendor (pulled out of the same bin), probably not. It sounds like multiple samples from the same bin only resulted from revisiting the same vendor, but the Methods need to be more explicit about that.

I see no statements about the sequences having been deposited in GenBank.

Reviewer #2: Overivew: The study by Korzik et al. describes the first investigation of shrimp mislabeling or species substitution in North Carolina. The study was apparently conducted over several semesters, spring and fall, as part of undergraduate training in DNA forensics. The authors point out that shrimp are an important and valuable North Carolina fishery and that mislabeling may result in mistrust of the local product. The authors collected and successfully analyzed to the species level via DNA barcoding 106 shrimp products from 60 retail vendors located in coastal and inland counties that were certified sold as “certified” “local” shrimp. Ten of the vendors were sampled more than one time. Results of the DNA species identity testing revealed that roughly one third of the products and vendors sold imported farmed shrimp as local wild-caught shrimp, remarkedly consistent with results from a previous nationwide study on shrimp mislabeling from 2014. Furthermore, there was no statistically significant difference between the mislabeling rates between inland and coastal counties. The results of repeat testing of vendors were unclear as presented. The study also showed that mislabeled products were sold at a cheaper price compared to honestly labeled products, but this result was not discussed.

The study lacks a description of statistical methods used and a Conclusion section. It could also use another table or graph to explain the repeated sampling of some vendors. The authors should note and cite other shrimp mislabeling studies to put their findings in context and expand on what the NC Catch certification entails. However, with these and other revisions noted below, this study should expand our sparse knowledge of shrimp mislabeling and its geographic reach, particularly in a state not previously investigated, yet known for its shrimp fishery. The study also nicely outlines the cultural and economic value of the shrimp fishery in NC and the current struggles the domestic shrimp industry faces.

Specific comments:

Abstract:

Line 27: As written, implies that that all the substituted whiteleg shrimp were from the eastern Pacific, when in fact they are farmed all over the world. Please clarify that the aquacultured species is native to that region.

Line 28: What is the negative ecosystem consequence of seafood importation? Please clarify or reword.

Introduction:

58: Please explain in more detail what the NC Catch certification entails. Does the advocacy organization spot check supply records or have other means of verifying that vendors are selling local products? This might be a topic for the Conclusion section.

63: “Mislabeling” of seafood could encompass misstating the country of origin or correct weight. Please larify that you are defining it as species substitution.

70-71: The Warner et al. 2014 citation concerns shrimp mislabeling only, not “seafood” in general, while the Warner et al. 2016 is correctly cited as referring to all seafood. Please clarify.

72-74. Please expand with examples or explain how mislabeling leads to the negative consequences described, such as human rights abuses and overfishing.

74-75: I believe that only testing or traceability along the entire seafood supply chain can pinpoint the source of mislabeling. Consider revising this sentence to clarify.

80-81: The authors should cite the other shrimp studies conducted.

Materials and Methods:

97-98: Describe whether shrimp were collected during the shrimp fishery season in NC, since the authors speculate later that this may be a factor in mislabeling (e.g. line 192).

133: Add a small section on statistical methods used

Results:

140: Reword “It is common farmed”. L. vannamei is the most commonly farmed shrimp species worldwide.

143-144: The sentence starting with “Of the ten..” is unclear. Consider representing the repeat testing of vendors in a table or graph to help the reader understand the results of this testing.

Discussion:

It would help to discuss the price difference observed between correctly labeled and mislabeled shrimp. Was the lower cost of mislabeled shrimp close to, higher or lower than the average price of farmed shrimp, sold under the correct label of farmed shrimp in NC?

171-173: Sentence starting with “Shrimp farms…” needs a reference.

177: Consider adding FDA import refusal data for shrimp as an updated reference for this section and discuss.

191-192: Begs the question if any sampling occurred when the shrimp fishery was closed or out of season. Please clarify here and in the Methods section.

195: Consider adding a Conclusion section on how to improve the NC catch certification and discuss whether a lower price for local shrimp may be “too good to be true” and an indication of potential mislabeling. For example, after the prevalent grouper mislabeling incidents in Florida, the state responded in marketing that conveyed what an honestly labeled grouper dish should cost. See: https://www.fdacs.gov/Food-Nutrition/Food-Safety-Resources/Mislabeling-Seafood-Products-Is-Illegal

6. PLOS authors have the option to publish the peer review history of their article (what does this mean?). If published, this will include your full peer review and any attached files.

Reviewer #1: No

Reviewer #2: No

---

## [Editor Report · Decision Letter 1]

10 Feb 2020

Marketplace Shrimp Mislabeling in North Carolina

PONE-D-19-24180R1

Dear Dr. Bruno,

We are pleased to inform you that your manuscript has been judged scientifically suitable for publication and will be formally accepted for publication once it complies with all outstanding technical requirements.

With kind regards,

Heather M. Patterson, Ph.D.

Academic Editor

PLOS ONE
---

## [Editor Report · Acceptance letter]

20 Feb 2020

PONE-D-19-24180R1 

Marketplace Shrimp Mislabeling in North Carolina 

Dear Dr. Bruno:

I am pleased to inform you that your manuscript has been deemed suitable for publication in PLOS ONE. Congratulations! Your manuscript is now with our production department. 

With kind regards,

on behalf of

Dr. Heather M. Patterson 

Academic Editor

PLOS ONE